# Effect of Near-Liquidus Squeeze Casting Pressure on Microstructure and Mechanical Property of AZ91D Alloy Differential Support

**DOI:** 10.3390/ma16114020

**Published:** 2023-05-27

**Authors:** Chunfang Zhao, Guangquan Ma, Tong Liu, Maoliang Hu, Zesheng Ji

**Affiliations:** 1School of Mechanical and Electrical Engineering, Lishui Vovational and Technical College, Lishui 323000, China; zhaochunfang0710@126.com; 2School of Materials Science and Chemical Engineering, Harbin University of Science and Technology, Harbin 150001, China; liutong_hrbust@163.com

**Keywords:** near-liquidus squeeze casting, AZ91D alloy, pressure, microstructure, mechanical property

## Abstract

In this study, near-liquidus squeeze casting AZ91D alloy was used to prepare differential support, and the microstructure and mechanical behavior under different applied pressure were investigated. Under the preset temperature, speed, and other process parameters, the effect of applied pressure on the microstructure and properties of formed parts was analyzed, and relevant mechanism was also discussed. The results showed that the ultimate tensile strength (UTS) and elongation (EL) of differential support can be improved by controlling real-time precision of the forming pressure. The dislocation density in the primary phase increased obviously with the pressure increasing from 80 MPa to 170 MPa, and even tangles appeared. When the applied pressure increased from 80 MPa to 140 MPa, the α-Mg grains were gradually refined, and the microstructure changed from rosette to globular shape. With increasing the applied pressure to 170 MPa, the grain could not be further refined. Similarly, its UTS and EL gradually increased with the applied pressure increasing from 80 MPa to 140 MPa. With increasing to 170 MPa, the UTS tended to be constant, but the EL gradually decreased. In other words, the UTS (229.2 MPa) and EL (3.43%) of the alloy reached the maximum when the applied pressure was 140 MPa, and the comprehensive mechanical properties were the best.

## 1. Introduction

Squeeze casting has become a hot research topic because of its advantages of good product quality and high production efficiency. Especially its applications in automobile parts have received widespread attention. It has developed aluminum alloy brackets, steering knuckles, and other load-bearing parts and successfully applied them to high-end brand cars [1,2]. AZ91 alloy has good casting performance [3], easy forming [4,5], sound damping [6], and damping characteristics [7]. It is the most widely used casting magnesium alloy currently and is widely used in the automobile industry. Czerwinski [8] carried out high-temperature die casting and near-liquidus temperature injection forming for AZ91D and AM60B magnesium alloys, which confirmed the feasibility of near-liquidus temperature injection forming in the near-liquidus casting of magnesium alloys. However, the application in magnesium alloys is limited due to the sizeable burning loss during the forming of magnesium alloys [9]. Mehr [10] applied near-liquidus casting technology to the semi-solid casting of AZ91D alloy and confirmed that the solidification rate could improve the dendrite size. Wang [11] found that using the low superheat casting process, AZ91D alloy with fine grain and non-dendritic structure was obtained when casting near 595 °C. Wang [2] proposed a new method of using low-frequency electromagnetic stirring to assist in near-liquidus squeeze casting of pure magnesium castings, refining the grain size of conventional casting castings from 10 mm to 232 µm. You [12] prepared semi-solid AZ91D magnesium alloy billets using the near liquidus insulation method. Implementing squeeze casting at 575 °C and T6 treated, the ultimate tensile strength, hardness, and elongation of AZ91D alloy reach the maximum values, with values of 285 MPa, 106.8 HV, and 13.36%, respectively. Inspired by the above research, this study attempts to control the pouring temperature of magnesium alloy close to the liquidus when implementing squeeze casting. Accordingly, a new idea of near-liquidus squeeze casting (NLSC) forming appeared. NLSC forming is used to maximize the advantages of the squeeze casting process that can continuously pressurize during solidification and obtain castings with excellent performance [13]. 

With the benefits of negligible porosity, strong mechanical qualities, exceptional surface smoothness, and precise dimensional accuracy, squeeze casting—in which liquid metal solidifies under pressure—is frequently used in the production of components. Numerous research studies have examined solidification pressure’s impact on microstructure and mechanical performance [14,15]. According to the findings, dendritic grains were produced by pressing casting at a high pouring temperature [16]. However, there is not much research done at temperatures close to liquidus, especially exploring the influence of pressure on the microstructure and mechanical properties of alloys under this condition. Based on the advantages of near-liquidus and squeeze casting, AZ91D alloy was used to fabricate the automobile differential support as the research object. This study mainly investigated the effect of applied pressure on the microstructure and mechanical properties during NLSC forming of AZ91D alloy. This method makes producing a suitable semi-solid microstructure possible without creating semi-solid blanks or slurries or keeping the metal solution in the holding furnace. Following NLSC forming, the spherical structure is more even and refined, enhancing its mechanical characteristics. Moreover, the study and development of this method support the advancement of the use and commercial production of squeeze-casting magnesium alloy. In the context of the entire industrial supply chain, it is crucial to support the creation and use of magnesium alloy and energy conservation and emission reduction to create a green cycle and sustainable development.

## 2. Materials and Methods 

In this paper, the alloys were prepared by remelting commercial AZ91D alloy with the casting temperature of 740 °C in an intermediate frequency furnace, and their actual chemical composition is obtained by ICP-AES (Liman Prodigy xp, Hudson, NY, USA), as shown in Table 1. During the whole forming process, the alloys were prepared by melting under the protection of CO_2_ (99 vol%) and SF_6_ (1 vol%). These protective gases were controlled by the flow valve, mixed and dried, and then output to the surface of magnesium liquid with a copper nozzle.

The experiments were carried out on the SCH-350A indirect squeeze casting machine. The tensile specimens were selected from the side of the differential support and machined into standard tensile specimens with a thickness of 3 mm, as shown in Figure 1. The tensile test was performed on a WDW-200 universal testing machine, and the fracture shape was observed by scanning electron microscopy. The samples were ground, polished, etched with 4% HNO_3_ alcohol solution (mass fraction), and observed microstructurally on an optical microscope. The thermal performance testing was prepared on a differential scanning calorimeter (DSC-404C, Netzch, Selb, Germany). In analyzing this research, the solid phase rate and the average size of solid phase particles in the semi-solid microstructure of the alloy were counted separately using mathematical software and subsequently calculated using the equation as follows [17]:(1)$$d_{0}=\left[\sum {2(A_{0}/\pi )}^{\frac{1}{2}}\right]/n$$
(2)$$f_{0}=(\sum P_{0}^{2}/4\pi A_{0})/n$$

To examine how pressure affects the temperature of AZ91D alloy, the solidification temperature of magnesium alloy during NLSC forming is measured in the way and its position is shown in Figure 2a. A K-type thermocouple was mounted on the stator core of the mold to measure the forming temperature, as shown in Figure 2b.

## 3. Results 

### 3.1. Selection of Pressure

Compared with gravity casting and high-pressure casting, the main advantage of squeeze casting is that the punch can continuously exert pressure on the solidified liquid metal in the mold cavity. At the same time, the pressure acting on the metal melt can not only refine the grain of the AZ91D alloy but also promote the partial plastic deformation of the melt, thus improving the performance of the parts [18]. In this test, the maximum operating pressure of the equipment is 180 MPa, and the selected pressure parameters are shown in Table 2.

### 3.2. Effect of Applied Pressure on Formability

Figure 3 shows the macroscopic morphology of the automotive differential support prepared by AZ91D alloy under the conditions of Table 2. It can be seen that the integrity of the differential support increases slightly with the pressure increases. This was because pressure mainly played the role of complementary shrinkage and solidification at a later stage under pressure during the squeeze casting filling [19]. When the melt temperature was low and the pressure injection speed was slow, the magnesium alloy melt fluidity was poor, and the complementary shrinkage effect was not apparent.

### 3.3. Effect of Applied Pressure on Microstructure

Figure 4 shows the microstructure of the automotive differential support at the casting temperature of 605 °C and the extrusion speed of 0.15 m/s under different applied pressures. As shown in Figure 4, the microstructure mainly consists of rose and broken spherical crystals when the externally applied pressure is 80 MPa. When the externally applied pressure is 110 MPa, the rose-like crystals disappear, but rose-petal-like and spherical crystals replace them. When the pressure increases to 140 MPa, the rosette-like crystals disappear. The microstructure is mainly dominated by fine and uniform spherical crystals. When the pressure was further increased to 170 MPa, the microstructure did not change much compared with that at 140 MPa. Thus, the microstructure inside the NLSC molded parts was transformed from rosy crystals to spherical crystals with the increase in applied pressure, and the critical pressure for the formation of spherical crystals was 140 MPa. In addition, the subsequent increase in pressure has little effect on the microstructure.

Figure 5 shows the grain size and spherical coefficient of the AZ91D alloy after NLSC forming under different pressure. Their average grain size tends to decrease and then increase as the pressure increases, but the average spherical coefficient tends to increase and then decrease. When the pressure was 140 MPa, the average grain size was 21.83 μm, and the average spherical coefficient was 0.59. But when the pressure was increased to 170 MPa, the average grain size increased to 23.46 μm, and the average spherical coefficient decreased slightly to 0.57. The main reason for this is that when pressure is more than critical pressure, the refined grain effect will decline and promote grain growth.

Based on the previous results of the authors, the second phase in AZ91D alloy was dominated by β-Mg_17_Al_12_ [20]. To further explore the influence of pressure on the second phase of AZ91D alloy during NLSC forming, SEM analysis was carried out, as shown in Figure 6. It can be seen that the applied pressure has a noticeable effect on the content and distribution of the second phase.

To further analyze the effect of applied pressure on the volume fraction of the second phase inside AZ91D alloy, the statistical results in Figure 6 is shown in Figure 7. Under 80 MPa, the content of the β-Mg_17_Al_12_ phase within the alloy was 30.94%, which was not uniformly distributed and has an aggregation phenomenon. When the pressure increased to 110 MPa, the content of the β-Mg_17_Al_12_ phase decreases, the size also became smaller, and its distribution was not uniform. When the applied pressure increased to 140 MPa, the content and size of the β-Mg_17_Al_12_ phase decreased to the lowest. When the pressure further increased to 170 MPa, the second phase content and grain size tended to increase compared with the pressure of 140 MPa, and the distribution of the second phase was also more inhomogeneous. It is concluded that the applied pressure limits the nucleation and growth of the β-Mg_17_Al_12_ phase within AZ91D alloy, which makes the content and size of the β-Mg_17_Al_12_ phase decrease with the increase in pressure. When the pressure reaches a specific value, the influence on the content and size of the second phase is very small.

Figure 8 shows the TEM results of the microstructure inside the primary phase of AZ91D alloy after applying different applied pressures. It can be seen from the figure that there are apparent dislocations inside the primary α-Mg grains when the applied pressure is 80 MPa. The number of dislocations is small, and the spacing of dislocations is large. When the applied pressure increased to 110 MPa, the dislocation density inside the primary phase increased significantly, but the dislocations remained isolated. With the further increase in the applied external pressure to 140 MPa, the dislocation density in the primary α-Mg grains increased further, and a few dislocations became intertwined. When the external pressure applied to the melt reaches 170 MPa, the dislocation density inside the alloy does not change significantly compared to 140 MPa. Nevertheless, the inter-dislocation entanglement phenomenon becomes more prominent. According to the solidification characteristics of AZ91D alloy during NLSC forming, it is known that the applied pressure has a significant effect on the morphology of dislocations inside the primary α-Mg grains. In the early stage of melt solidification, the alloy is mainly in the liquid phase, and no apparent dislocations are produced inside the alloy. As the melt temperature decreased, the solid phase content increased. Under the continuous effect of the applied pressure, the crystal surface produced steps or other thermal deformation and dislocations. As the pressure applied externally increases, the number of dislocations generated within the alloy also increased. When the dislocation density increased to a certain degree, the probability of interactions between dislocations increased, and eventually, inter-dislocation entanglement occurred. Therefore, the elongation was affected. Wang [21] studied the GW103K alloy, and similar results appeared.

### 3.4. Effect of Pressure on Mechanical Properties and Fracture Morphology

The change in applied pressure during the NLSC forming of AZ91 alloy cause a microstructure change of the alloy, which inevitably caused a change in mechanical properties. Figure 9 shows the tensile curves of the prepared alloy at different applied pressures. Both the UTS and EL of the alloy showed a trend of first increasing and then decreasing. With the gradual increase in the applied pressure, the UTS (229.2 MPa) and EL (3.43%) reached the maximum value when the applied pressure was 140 MPa. Summarily, the adequately applied pressure during the NLSC forming is beneficial to the microstructure refinement, which improves the mechanical properties. Still, the elongation decreases when the pressure exceeds a specific critical value.

Figure 10 shows tensile fracture of AZ91D alloy under different applied pressures at room temperature. The fracture of alloy has a large size cleavage plane when forming pressure is 80 MPa. Meanwhile, there are apparent tearing edges near the deconstruction surface. When the pressure reaches 110 MPa, the cleavage plane of the alloy fracture shows a stepped shape, but the number is also more than that of 80 MPa, the fracture surface of the alloy shows a stepped shape, and the number of tearing edges is also more than that at 80 MPa. When the pressure increases to 140 MPa, the number of tearing edges at the fracture increases significantly, and there are many tough nests in addition to tearing edges. When the pressure reaches 170 MPa, the fracture site is mainly laminated and tearing edges. As the applied pressure increases during NLSC forming, the area of the decoupling surface at the room temperature tensile fracture of the alloy gradually decreases. And the number of tearing edges and tough nests increases significantly, which eventually leads to the enhancement of the toughness and plasticity of the alloy. The best overall performance of the forming parts was achieved when the applied pressure of NLSC forming was 140 MPa.

## 4. Discussion

### 4.1. Effect of Pressure on Solidification Temperature

Metal solidification is the metal transformation process from liquid state to solid state, mainly consisting of two processes: nucleation and growth of grains. Based on the thermodynamic point, the solidification of metals will be influenced by heat flow and pressure [22]. The current metal solidification theory was developed in gravity casting, which only considered the flow of molten metal at high temperatures without considering the influence of external stress on it. However, with the increasing number of new casting methods, the factors affecting the solidification process of alloys are becoming more and more complex. According to a relevant study, applying pressure during NLSC formation may change the solidification behavior of the alloy [23,24]. Hence, adjusting the applied pressure might affect its microstructure and mechanical properties. It is generally believed that the applied pressure during the solidification of an alloy affects the phase transition, which can be derived from the Clausius–Clapeyron equation as follows [25]:(3)$$\frac{\Delta T_{f}}{\Delta P}=\frac{T_{f}\left(V_{l}-V_{s}\right)}{\Delta H_{f}}$$
where $T_{f}$ is the solidification temperature of the alloy in the equilibrium state, $V_{l}$ is the specific volume in the liquid state, $V_{s}$ is the specific volume in the solid state, and $\Delta H_{f}$ is the latent heat of melting.

Substitute volume in Equation (3), and the effect of pressure on the solidification point of alloy can be simplified as follows:(4)$$P=P_{0}\exp\left(\frac{-\Delta H_{f}}{RT_{f}}\right)$$
where $P_{0}$ is the standard atmospheric pressure, $\Delta H_{f}$ is a constant, and R is a constant.

From Equation (4), we can conclude that the solidification temperature ($T_{f}$) in the equilibrium state will increase with the increase in externally applied pressure (*P*). The solidification phase diagram of the alloy will also change, i.e., the liquidus temperature of the alloy will increase. Meanwhile, the eutectic point shifted to the upper left corner. For most alloys, the rise in liquidus temperature generated during solidification is about 10^−2^ K/atm [26]. Increase in the $T_{f}$ value means that the sub-cooling of the alloy melt during solidification under pressure rises significantly, and the increase in sub-cooling can effectively promotes the nucleation of the alloy melt, which has a strong effect on promoting the grain refinement of the cast alloy [27].

To obtain the appropriate temperature for the heat treatment, differentia scanning calorimetry of the prepared specimens was performed using a (DSC-404C, Netzch, Germany) differential scanning calorimeter, and the test results are shown in Figure 11a. The curves show that the alloy produced two heat absorption peaks during continuous heating, corresponding to 436.928 °C and 595.728 °C. Therefore, the liquidus temperature of AZ91D magnesium alloy can be determined as 595.728 °C. For magnesium alloys, the primary α-Mg phase within the AZ91D alloy generally formed at the liquidus temperature. It can be determined by the onset temperature of the primary α-Mg phase growth. Figure 11 and Table 3 show the solidification temperature of AZ91D alloy at different applied pressures, i.e., its liquidus temperature (black line). One can obtain the following relationship by fitting the onset temperature of the formation of the primary α-Mg phase in AZ91D alloy to the pressure.
(5)$$T_{L}=0.066P+598.350$$

Equation (5) shows that the onset of formation of the primary α-Mg phase at atmospheric pressure is 598.357 °C, an increase of 2.629 °C compared with the liquidus temperature of AZ91D alloy at atmospheric pressure of 595.728 °C. Because the formation of the primary α-Mg phase starts at the liquidus temperature, this can be considered a systematic error of the test and can be corrected by taking −2.629 °C as a correction value to Equation (3). The revised equation for the relationship between liquid-phase temperature and pressure for AZ91D alloy is obtained, as shown in the red line in Figure 11.
(6)$$T_{L}^{*}=0.066P+595.721$$
where $T_{L}^{*}$ is the solidification temperature of AZ91D alloy.

By comparing the test-measured results at different applied pressures with the results obtained from the calculation of Equation (6), it can be seen that the liquidus temperature of the alloy at different applied pressures is close to each other, which is also closer to the conclusion of the literature [23]. The calculated results are shown in Table 4. The liquidus temperature will increase by 0.066 °C for every 1 MPa increase in pressure during the NLSC forming of AZ91D alloy. At a pressure of 140 MPa, it will cause the liquidus temperature to rise by 9.233 °C compared to that at atmospheric pressure. The rise in the liquidus of the alloy in the figure means that the sub-cooling of the alloy melt increases when it solidifies under pressure, and the increase in sub-cooling effectively promotes homogeneous nucleation of the alloy melt. The increase in sub-cooling is the primary source of homogeneous nucleation, so the increase in the liquidus strongly affects the refinement of the as-cast alloy.

### 4.2. Effect of Pressure on Solidification Behavior

It is well-known that the casting temperature significantly affected the internal grain size of the molded parts. During NLSC forming of AZ91D alloy, a pressure is set to 110 MPa and a press injection speed is set to 0.15 m/s. Accordingly, decreasing the pouring temperature could make the grains refine, improve the grain spherical coefficient, and improve the homogenization of the β-Mg_17_Al_12_ phase at grain boundaries. Table 5 shows the relationship between the pouring temperature and the liquidus temperature of AZ91D alloy at a forming pressure of 110 MPa. It can be seen that 595 °C is used as the pouring temperature. However, this temperature is on the liquidus at atmospheric pressure. The liquidus of the alloy has risen to 602.981 °C due to the external applied pressure of 110 MPa. Meanwhile, the pouring temperature of 595 °C is nearly 8 °C below the liquidus, and the alloy is in a semi-solid state.

According to the above analysis results, the grain size decreases first but then increases with the applied pressure when AZ91D alloy is NLSC formed at the pouring temperature of 605 °C. The grain of alloy is the finest at 140 MPa. However, the grain size, the distribution of second phase, and the spherical coefficient all changed between 140 MPa and 170 MPa. The relationship between liquidus temperature and pouring temperature of AZ91D alloy under different pressures was compared and analyzed to investigate how the pressure works, as shown in Table 6.

By comparing the above results, it is found that when the applied pressure is 80 MPa, 110 MPa, and 140 MPa, respectively, the pouring temperature of the alloy is higher than its liquidus temperature. When the applied pressure increases to 170 MPa, although the forming temperature (605 °C) is higher than the liquidus of atmospheric casting (595.728 °C), this temperature is nearly 2 °C lower than the liquidus of 170 MPa. As a result, the grain size of AZ91D alloy is more coarse than that of 140 MPa. At the same time, the coarsening of grains also worsens the mechanical properties of alloy.

### 4.3. Establishment of Pressure Fusing and Spheroidizing Dendrite Model

When a metal is solidifying, if the radius of the interfacial surface at the solid–liquid phase interface is assumed as *r*, then the modification of its Gibbs free energy is represented by Equation (7):(7)$$\Delta G=\sigma_{SL}\Delta A=\sigma_{SL}\Delta Vk=2\sigma_{SL}\Delta V\frac{1}{r}$$
where $\sigma_{SL}$ is the interfacial surface at the solid–liquid phase interface, ΔV is the molar volume, *k* is the average curvature of the interface of a curved body.

For a curved body, its average curvature *k* can be expressed by Equation (8) [28]:(8)$$k=\frac{\Delta A}{\Delta V}=\frac{1}{r_{1}}+\frac{1}{r_{2}}$$
where $r_{1}$ and $r_{2}$ are the radius of curvature of any two arcs on the surface. For a sphere, $r_{1}=r_{2}=r$, then k=2/r.

Generally, the surface has higher free energy than the plane, and this higher free energy will cause the melting point to drop:(9)$$\Delta T=T_{CR}-T_{R}$$
where $T_{CR}$ is the melting point of solid particles when the interface is planar, and $T_{R}$ is the melting point of solid particles when the interface is curved.

When ΔT is small, the Gibbs free energy of the alloy can be calculated by Equation (10) [29]:(10)$$\Delta G=\Delta H-T_{R}\Delta S=\Delta H-T_{R}\frac{\Delta H}{T_{CR}}=\frac{\Delta H(T_{CR}-T_{R})}{T_{CR}}=\frac{\Delta H}{T_{CR}}\Delta T$$
where ΔH is the transformation of enthalpy during solid–liquid transformation, and ΔS is the transformation of entropy during solid–liquid transformation.

Equations (11) and (12) can be obtained from Equations (7) and (10):(11)$$2\sigma_{SL}\Delta V\frac{1}{r}=\frac{\Delta H}{T_{CR}}\Delta T$$
(12)$$\Delta T=\frac{2\sigma_{SL}\Delta VT_{CR}}{r\Delta H}$$

By substituting Equation (9) into Equations (11) and (12), Equations (13) and (14) can be obtained.
(13)$$T_{CR}-T_{R}=\frac{2\sigma_{SL}\Delta VT_{CR}}{r\Delta H}$$
(14)$$T_{R}=T_{CR}-\frac{2\sigma_{SL}\Delta VT_{CR}}{r\Delta H}$$

Generally, the solid–liquid interface energy of alloy will change under pressure, as shown in Equation (15):(15)$$\sigma_{SL}=\sigma_{SL}^{*}+rT_{CR}\ln P$$
where, $\sigma_{SL}^{*}$ is the solid–liquid interface energy under no pressure.

Substituting Equation (15) into Equation (14), Equation (16) can be obtained as
(16)$$T_{R}=T_{CR}-\frac{2\Delta VT_{CR}(\sigma_{SL}^{*}+rT_{CR}\ln P)}{r\Delta H}=T_{CR}-\frac{2\sigma_{SL}^{*}\Delta VT_{CR}}{\Delta H}\cdot \frac{1}{r}-\frac{2\Delta VT_{{}_{CR}}^{2}}{\Delta H}\cdot \ln P$$

From Equation (16), a change in surface radius will have an effect on the surface equilibrium melting point ($T_{R}$). In particular, the smaller the surface radius (*r*) and the greater the curvature (*k*), the lower the surface equilibrium melting point. In general, the equilibrium melting point of the concave and convex parts of the same surface will be lower because the radius of the surface is smaller, and the curvature is more significant. When the surrounding metal liquid temperature is higher, these parts will be melted into the liquid phase, so that the curvature of the part gradually becomes smaller, as the concave part is fused, and the convex part is rounded continuously. From the third term on the right side of Equation (16), the pressure affects the equilibrium melting point of the surface. The higher the pressure, the lower the equilibrium melting point of the surface. Therefore, the applied pressure lowers the equilibrium melting point of the concave and convex parts, which creates favorable conditions for the fusion and rounding of the dendrites. In addition, the self-stirring turbulent flow generated by the metal liquid filling also accelerates the fracture of the rosettes and dendrite roots under mechanical pressure, and the fractured dendrites are further rounded, spheroidized, and become new cores to continue growing, thus refining the casting microstructure [30]. Figure 12 shows a model of the process of fusing dendrites under pressure to form new cores.

## 5. Conclusions

(1)With the increase in pressure from 80 MPa to 140 MPa, the microstructure of near-liquidus squeeze casting AZ91D alloy gradually evolves from rose-like crystals to spherical crystals, and the average grain size is refined from 35.07 μm to 21.83 μm, and the average spherical coefficient increases from 0.45 to 0.59. The pressure simultaneously reduces the content of the second phase β-Mg_17_Al_12_ within the alloy so that it gradually transforms from a coarse aggregated state to a fine homogeneous state.(2)With the increase in pressure from 80 MPa to 140 MPa, the tensile strength increases from 209.4 MPa to 229.2 MPa, and the elongation increases from 2.70% to 3.43%. When the pressure exceeded 140 MPa, the pressure had little effect on the microstructure and mechanical properties of the alloy, but the grain size was coarsened, and the mechanical properties were decreased.(3)During the NLSC forming process, the liquidus temperature of AZ91D alloy will increase by 0.066 °C for every 1 MPa increase in pressure. At 140 MPa pressure, it will cause the liquidus temperature of AZ91D alloy to increase by 9.233 °C compared with that at atmospheric pressure.(4)Under pressure, the AZ91D alloy melt solidifies into a nucleus, and the equilibrium melting point of the part with a smaller radius and more significant surface curvature decreases. The concave and convex parts of the grain surface are melted first, so the rosy crystal or dendrite crystal arm will be fused and rounded, and the self-stirring turbulence generated by the pressure in the metal liquid filling accelerates this fusion and fracture. The rounded dendrite becomes a new nucleus. The core continues to grow, thus refining the microstructure of the casting.

## Figures and Tables

**Figure 1 materials-16-04020-f001:**
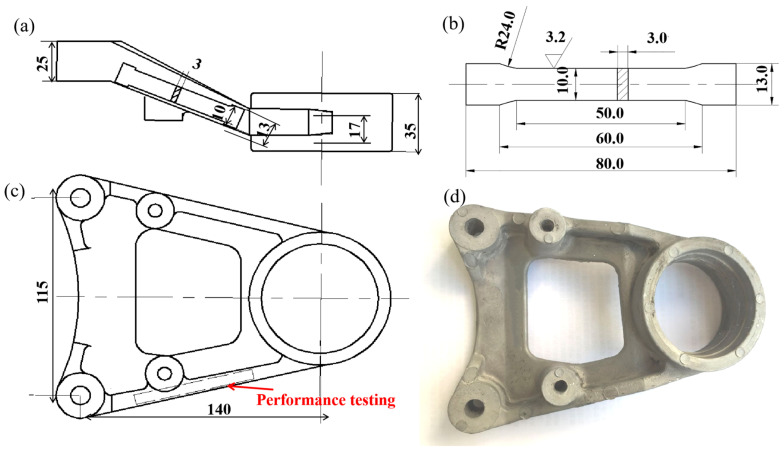
Pictures of machined parts and their related parameters. (**a**) Front view; (**b**) tensile specimen size; (**c**) vertical view; (**d**) physical picture.

**Figure 2 materials-16-04020-f002:**
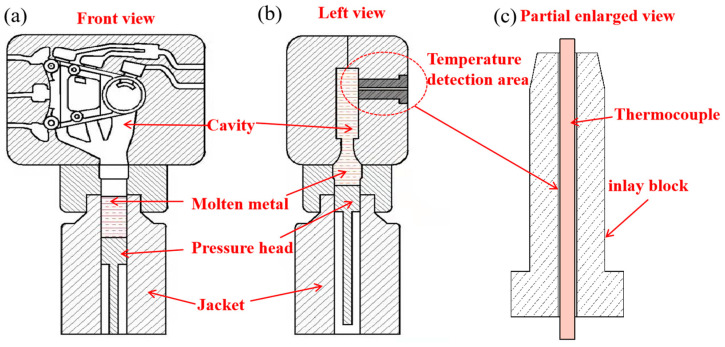
The structure of pouring row and the position of the thermocouple in the mold.

**Figure 3 materials-16-04020-f003:**
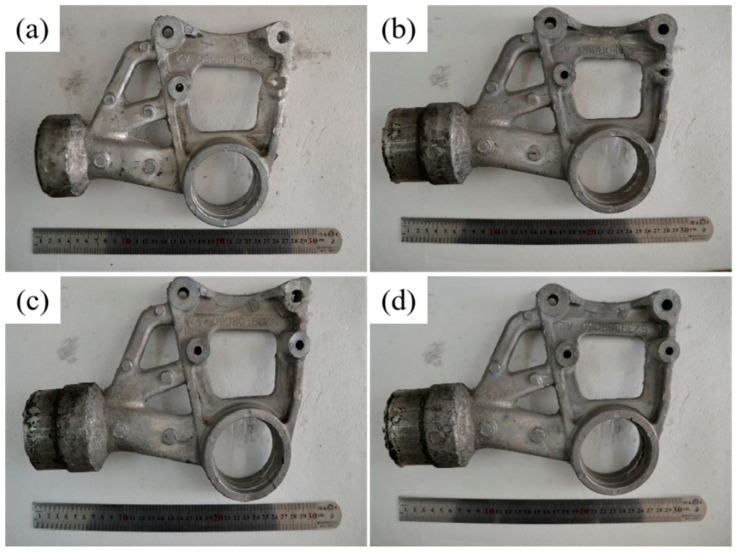
Pictures of automobile differential support prepared under different applied pressures. (**a**) 80 MPa; (**b**) 110 MPa; (**c**) 140 MPa; (**d**) 170 MPa.

**Figure 4 materials-16-04020-f004:**
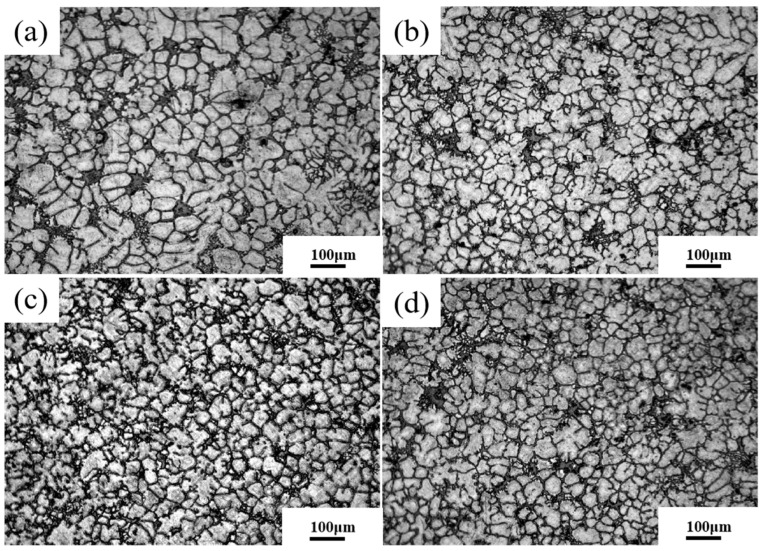
Microstructures of differential support at different applied pressures by NLSC. (**a**) 80 MPa; (**b**) 110 MPa; (**c**) 140 MPa; (**d**) 170 MPa.

**Figure 5 materials-16-04020-f005:**
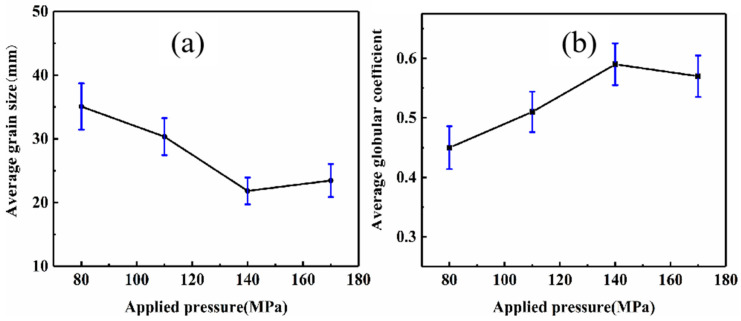
Effect of pressure on average grain size (**a**) and globular coefficient (**b**) of AZ91D alloy.

**Figure 6 materials-16-04020-f006:**
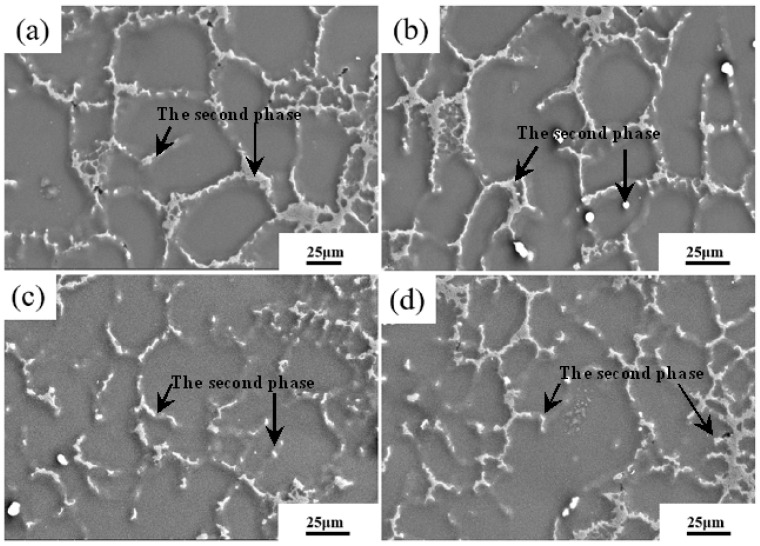
SEM micrographs under different applied pressures. (**a**) 80 MPa; (**b**)110 MPa; (**c**) 140 MPa; (**d**) 170 MPa.

**Figure 7 materials-16-04020-f007:**
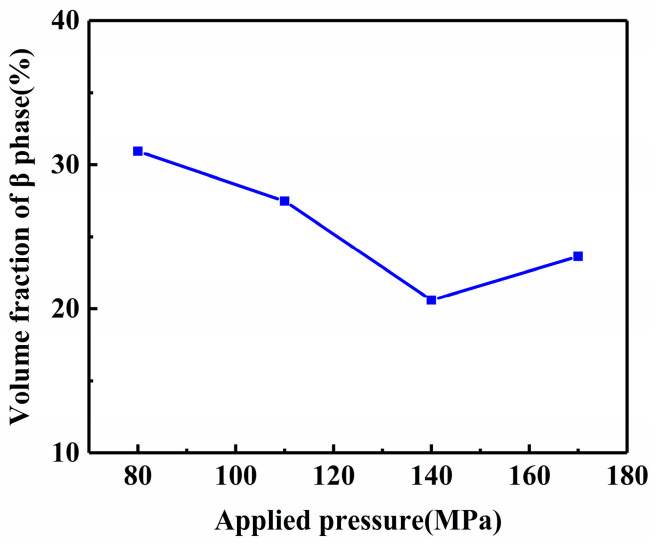
Volume fraction of β-Mg_17_Al_12_ phase under different applied pressures.

**Figure 8 materials-16-04020-f008:**
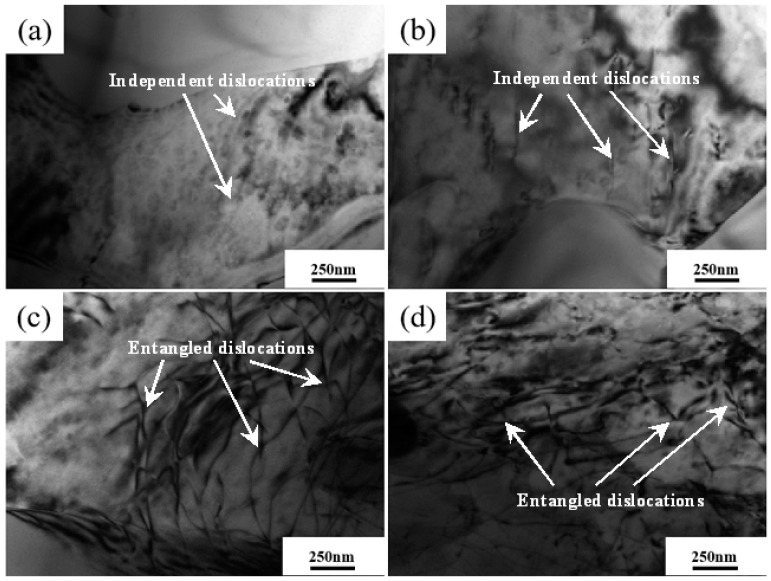
Internal microstructure of primary phase of AZ91D alloy under different pressures. (**a**) 80 MPa; (**b**) 110 MPa; (**c**) 140 MPa; (**d**) 170 MPa.

**Figure 9 materials-16-04020-f009:**
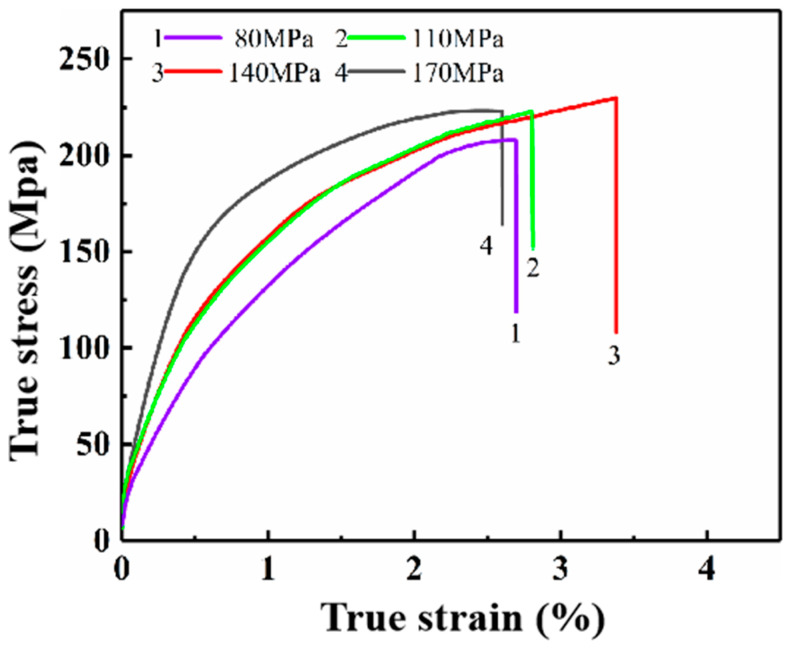
Effect of pressure on tensile strength and elongation of differential support.

**Figure 10 materials-16-04020-f010:**
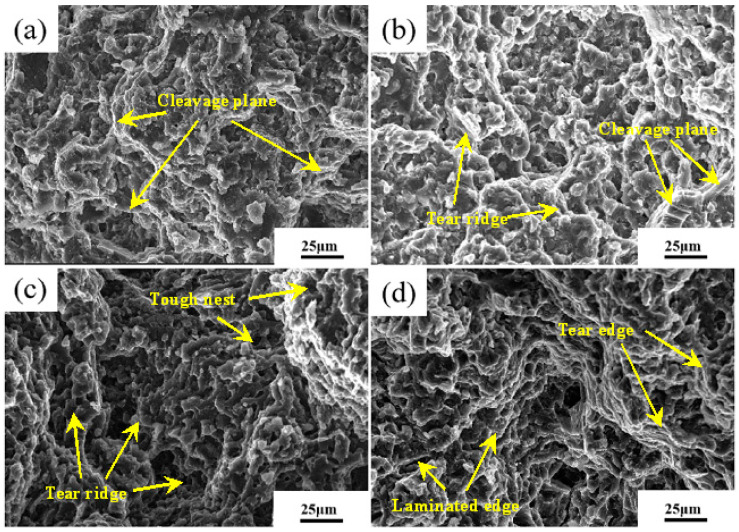
Effect of pressure on tensile fracture morphology of differential support by NLSC. (**a**) 80 MPa; (**b**) 110 MPa; (**c**) 140 MPa; (**d**) 170 MPa.

**Figure 11 materials-16-04020-f011:**
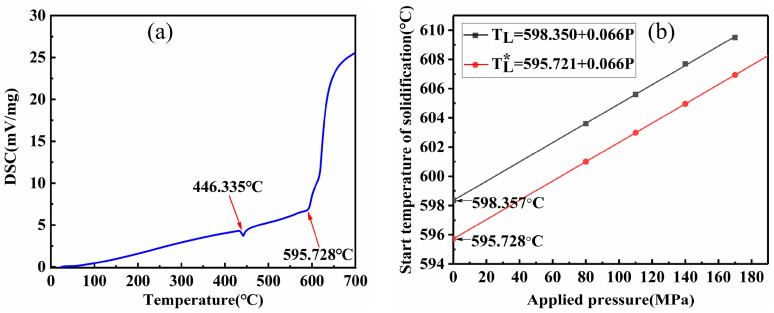
Thermal analysis of AZ91D alloy. (**a**) DSC curves, (**b**) initial solidification temperature.

**Figure 12 materials-16-04020-f012:**
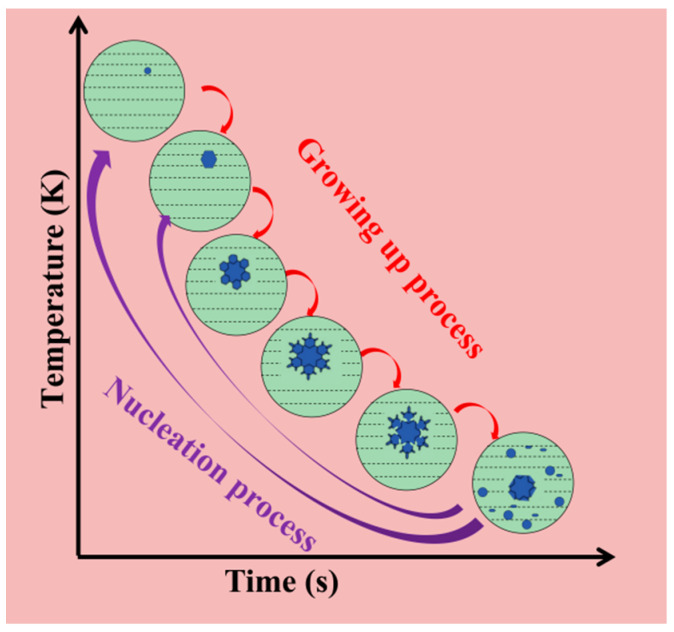
Process of dendrite fusing under pressure to form new crystal nuclei.

**Table 1 materials-16-04020-t001:** Chemical composition of AZ91D alloy (wt.%).

Al	Zn	Mn	Be	Si	Cu	Fe	Ni	Mg
8.959	0.684	0.203	0.001	0.022	0.002	0.003	0.001	Bal.

**Table 2 materials-16-04020-t002:** Testing parameters.

No.	Pouring Temperature (°C)	Applied Pressure (MPa)	Injection Velocity (m/s)	Mold Temperature (°C)	Holding Time (s)
1	605	80	0.15	300	20
2	605	110	0.15	300	20
3	605	140	0.15	300	20
4	605	170	0.15	300	20

**Table 3 materials-16-04020-t003:** Temperature of primary α-Mg formed under different applied pressures.

Applied Pressures (MPa)	Solidification Temperature (°C)
80	603.6
110	605.6
140	607.7
170	609.5

**Table 4 materials-16-04020-t004:** Liquidus of AZ91D alloy under different applied pressures.

Applied pressures (MPa)	0.101	80	110	140	170
Liquidus temperature (°C)	595.728	601.001	602.981	604.961	606.941
Liquidus temperature rise value (°C)	0	+5.273	+7.253	+9.233	+11.213

**Table 5 materials-16-04020-t005:** Difference between pouring temperature and liquidus temperature under 110 MPa.

Pouring temperature (°C)	595	605	615	625
Liquidus temperature at a pressure of 110 MPa (°C)	602.981	602.981	602.981	602.981
Difference between the above two (°C)	−7.981	+2.019	+12.019	+22.019

**Table 6 materials-16-04020-t006:** Difference between pouring temperature and liquidus temperature of AZ91D alloy under different applied pressures.

Applied pressure (MPa)	80	110	140	170
Pouring temperature (°C)	605	605	605	605
Liquidus temperature	601.001	602.981	604.961	606.941
Difference between the above two (°C)	+3.999	+2.019	+0.039	−1.941

## Data Availability

Data will be made available on request.

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
