# Peer review of "Effect of Near-Liquidus Squeeze Casting Pressure on Microstructure and Mechanical Property of AZ91D Alloy Differential Support"

_materials, 2023, doi:10.3390/ma16114020_

Round 1

Reviewer 1 Report

1)    The objective and novelty of the work to be mentioned at the end of Introduction section.

2)    The Introduction section is to be strengthened by incorporating more quality literature content. The theoretical concept to the research and similar findings if at all there in the past work to be included in the Introduction section.

3)    A number of locations there is format mistake like, 180MPa. Please give one-unit gap between the number and unit for format uniformity.

4)    On explaining microstructures of differential support at different applied pressures by NLSC, you mentioned that the critical pressure for the formation of spherical crystals was 140 MPa. What is the basis for mentioning the critical pressure for the formation of spherical crystals as 140 MPa? How the optical microstructure gives this information? Please explain.

5)    Explaining the Figure 6, it is stated that based on the previous results of the authors, the second phase in AZ91D alloy was dominated by β-Mg17Al12 [15]. It is from the literature. But the microstructure shows only one type of phase with well defined grain boundary. Also, it is mentioned as “applied pressure has a noticeable effect on the content and distribution of the second phase”. I cannot see any second phase in the microstructure. Please mark the second phase in the microstructure and explain accordingly.

6)    Under Figure 8, it is spoken about dislocation density and entangling of dislocations. Please mark the dislocations in the image itself. Also, the explanations given under this do not have any experimental proof. Please give atleast literature proof to the justification of your answer.

7)    Please give justification for your arguments for tensile fracture images (Figure 10). It is stated under the figure 10 that “The best overall performance of the forming parts was achieved when the applied pressure of NLSC forming was 140 MPa”. How you can pass the judgement like this from image comparison? Explain. Please add literature support where ever experimental support is weak.

8)    Reference number 18, the title of the paper, first letter of each word is of upper-case letter whereas in the remaining, only first letter of the 1st word is of upper-case letter. Please maintain format uniformity.

1)    The objective and novelty of the work to be mentioned at the end of Introduction section.

2)    The Introduction section is to be strengthened by incorporating more quality literature content. The theoretical concept to the research and similar findings if at all there in the past work to be included in the Introduction section.

3)    A number of locations there is format mistake like, 180MPa. Please give one-unit gap between the number and unit for format uniformity.

4)    On explaining microstructures of differential support at different applied pressures by NLSC, you mentioned that the critical pressure for the formation of spherical crystals was 140 MPa. What is the basis for mentioning the critical pressure for the formation of spherical crystals as 140 MPa? How the optical microstructure gives this information? Please explain.

5)    Explaining the Figure 6, it is stated that based on the previous results of the authors, the second phase in AZ91D alloy was dominated by β-Mg17Al12 [15]. It is from the literature. But the microstructure shows only one type of phase with well defined grain boundary. Also, it is mentioned as “applied pressure has a noticeable effect on the content and distribution of the second phase”. I cannot see any second phase in the microstructure. Please mark the second phase in the microstructure and explain accordingly.

6)    Under Figure 8, it is spoken about dislocation density and entangling of dislocations. Please mark the dislocations in the image itself. Also, the explanations given under this do not have any experimental proof. Please give atleast literature proof to the justification of your answer.

7)    Please give justification for your arguments for tensile fracture images (Figure 10). It is stated under the figure 10 that “The best overall performance of the forming parts was achieved when the applied pressure of NLSC forming was 140 MPa”. How you can pass the judgement like this from image comparison? Explain. Please add literature support where ever experimental support is weak.

8)    Reference number 18, the title of the paper, first letter of each word is of upper-case letter whereas in the remaining, only first letter of the 1st word is of upper-case letter. Please maintain format uniformity.

Author Response

1) The objective and novelty of the work to be mentioned at the end of Introduction section.

Response:Thank you for your guidance. We have added the purpose and novelty of this work in the introduction section and highlighted it in red.

“This method makes producing a suitable semi-solid microstructure possible without creating semi-solid blanks or slurries or keeping the metal solution in the holding furnace. Following NLSC forming, the spherical structure is more even and refined, enhancing its mechanical characteristics. Moreover, the study and development of this method support the advancement of the use and commercial production of squeeze-casting magnesium alloy. In the context of the entire industrial supply chain, it is crucial to support the creation and use of magnesium alloy and energy conservation and emission reduction to create a green cycle and sustainable development.”

2) The Introduction section is to be strengthened by incorporating more quality literature content. The theoretical concept to the research and similar findings if at all there in the past work to be included in the Introduction section.

Response:Thank you very much for your reminder, which is very important for revising our manuscript. For this problem, I have corrected it in the manuscript and highlighted it in red. The main referential literature comprising the above-mentioned content in relevance is listed.

“Wang[12] proposed a new method of using low-frequency electromagnetic stirring to assist in near-liquidus squeeze casting of pure magnesium castings, refining the grain size of conventional casting castings from 10mm to 232 µm. You[13] prepared semi-solid AZ91D magnesium alloy billets using the near liquidus insulation method. Implementing squeeze casting at 575 °C and T6 treated, the ultimate tensile strength, hardness, and elongation of AZ91D alloy reach the maximum values, with values of 285MPa, 106.8HV, and 13.36%, respectively.”

“With the benefits of negligible porosity, strong mechanical qualities, exceptional surface smoothness, and precise dimensional accuracy, squeeze casting-in which liquid metal solidifies under pressure-is frequently used in the production of components. Numerous research studies have examined solidification pressure's impact on microstructure and mechanical performance[15-16]. According to the findings, dendritic grains were produced by pressing casting at a high pouring temperature[17]. However, there is not much research done at temperatures close to liquidus, especially exploring the influence of pressure on the microstructure and mechanical properties of alloys under this condition.”

3) A number of locations there is format mistake like, 180MPa. Please give one-unit gap between the number and unit for format uniformity.

Response:I am very grateful for your professional opinion. Under the guidance of your professional opinions, we changed the gap between the number and the unit and highlighted it in red.

4) On explaining microstructures of differential support at different applied pressures by NLSC, you mentioned that the critical pressure for the formation of spherical crystals was 140 MPa. What is the basis for mentioning the critical pressure for the formation of spherical crystals as 140 MPa? How the optical microstructure gives this information? Please explain.

Response:I'm very grateful for your professional opinion. The method uses the normal formula for the solid phase rate and the average size of solid phase particles calculation, as seen in the red highlighted part in section 2.

5) Explaining the Figure 6, it is stated that based on the previous results of the authors, the second phase in AZ91D alloy was dominated by β-Mg17Al12 [15]. It is from the literature. But the microstructure shows only one type of phase with well defined grain boundary. Also, it is mentioned as “applied pressure has a noticeable effect on the content and distribution of the second phase”. I cannot see any second phase in the microstructure. Please mark the second phase in the microstructure and explain accordingly.

Response:I'm very grateful for your professional opinion. I have corrected this part in the revised manuscript and highlighted it in red, as shown in Figure 6.

6) Under Figure 8, it is spoken about dislocation density and entangling of dislocations. Please mark the dislocations in the image itself. Also, the explanations given under this do not have any experimental proof.

Response:Thank you very much for your careful review of our manuscript. To make the captions of those figures more comprehensive and accurate, I have corrected the captions of Fig. 8 in red.

7) Please give justification for your arguments for tensile fracture images (Figure 10). It is stated under the figure 10 that “The best overall performance of the forming parts was achieved when the applied pressure of NLSC forming was 140 MPa”. How you can pass the judgement like this from image comparison? Explain. Please add literature support where ever experimental support is weak.

Response:I'm very grateful for your professional opinion. As is well known, the strength and plasticity of metals are the two most important indicators for characterizing the mechanical properties of alloys. As shown in Figure 9, the alloy has maximal strength and elongation when the pressure reaches 140 MPa. Thus, we obtain the above conclusion

8) Reference number 18, the title of the paper, first letter of each word is of upper-case letter whereas in the remaining, only first letter of the 1st word is of upper-case letter. Please maintain format uniformity.

Response:I am very sorry for this mistake. Thanks again for your professional advice, the modified section is shown in reference 18.

Reviewer 2 Report

materials-2369998

Title: Effect of near-liquidus squeeze casting pressure on microstructure and mechanical property of AZ91D alloy differential support

The subject of the presented study is interesting, however, there are some raised issues that need to be treated.

Introduction:

1.    The introduction is very limited. Please provide some more details about the previous studies.

2.    Compacting more than two references such as [3-7] in one statement is not preferred. Please bring the specific detail of each reference.

Materials and methods

3.    What is the “ICP-AES” if this is the chemical analysis set, please indicate the model, manufacturer, city and country.

4.     Indicate what is the “SF6”?

5.    Please clear “the volume flow ratio of the two is about 99:1”

6.    Please refer to the tensile test specimen standard. Is the tensile specimen casted in a special mold or taken from a cast part?

Result

7.    ” but also promote the partial plastic deformation of the melt”, how it comes?

8.    Please clear, what is the spherical coefficient? How it is determined?

9.    On Y-axis of Figure 9, please simply it to “Stress” or “Engineering stress”.

10.Is the constant R is the general gas constant.

11.How did you evaluate the constants DH , and what is the effect of the pressure on it?

12.How can you detect the point of formation of the primary α-Mg phase, this may need using dilatometric investigation.

13.The difference in the liquidus point “2.629°C” not clear. Is this difference valid at the different pressure?

14.The basics of the suggested nucleation and spheredoizing model shown in Fig 12 is not clear.

Author Response

Introduction:

1. The introduction is very limited. Please provide some more details about the previous studies.

Response:Thank you very much for your reminder, which is very important for revising our manuscript. For this problem, I have corrected it in the manuscript and highlighted it in red. But beyond that, as a postscript, the main referential literature comprising the above-mentioned content in relevance is listed.

2. Compacting more than two references such as [3-7] in one statement is not preferred. Please bring the specific detail of each reference.

Response:To make those references more comprehensive and accurate, I have corrected them in the manuscript and highlighted it in red.

Materials and methods

3. What is the “ICP-AES” if this is the chemical analysis set, please indicate the model, manufacturer, city and country.

Response:I'm very grateful for your professional opinion. In the revised manuscript, we have added information related to the instrument model.

4. Indicate what is the “SF6”?

Response:Thank you for your guidance. We have submitted a new version of the manuscript with carefully polished language. We have addressed the comments raised by the reviewers, and the amendments are highlighted in red in the revised manuscript. We notably revised some sentences since they were not correct

5. Please clear “the volume flow ratio of the two is about 99:1”

Response:Thank you for your guidance. We have submitted a new version of the manuscript with carefully polished language. We have addressed the comments raised by the reviewers, and the amendments are highlighted in red in the revised manuscript. We notably revised some sentences since they were not correct

6. Please refer to the tensile test specimen standard. Is the tensile specimen casted in a special mold or taken from a cast part?

Response:Thank you for your careful revision of our manuscript and for your professional comments. The tensile specimen is taken from a cast part.

Result

7. ” but also promote the partial plastic deformation of the melt”, how it comes?

Response:Thank you for your careful revision of our manuscript and for your professional comments. I am sorry to make this mistake for my neglect, I marked the reference in the wrong position, this conclusion is obtained from previous literature.

8. Please clear, what is the spherical coefficient? How it is determined?

Response:I'm very grateful for your professional opinion. In the method, the normal formula for the solid phase rate and the average size of solid phase particles calculation is used; see the red highlighted part in section 2.

9. On Y-axis of Figure 9, please simply it to “Stress” or “Engineering stress”.

Response:Thank you for your guidance, I have corrected it.

10. Is the constant R is the general gas constant.

Response:Thank you very much for your careful review of our manuscript. In this manuscript, R is the general gas constant.

11. How did you evaluate the constants DH , and what is the effect of the pressure on it?

Response:Thanks to the reviewers for their suggestions. However, after careful examination, we found nothing related to "DH" in the text. Please provide us with more details. 

12. How can you detect the point of formation of the primary α-Mg phase, this may need using dilatometric investigation.

Response:Thank you for your professional advice. The phase transition temperature is determined by DSC measurements, the details are shown in Fig. 11 (a).

13.The difference in the liquidus point “2.629°C” not clear. Is this difference valid at the different pressure?

Response:The liquidus point “2.629°C” can be considered a systematic error of the test and can be corrected by taking -2.629°C, which is the same at the different pressure.

14. The basics of the suggested nucleation and spheredoizing model shown in Fig 12 is not clear

Response:Thank you very much for your professional feedback. In the revised manuscript, I have redrawn the schematic diagram, as shown in Figure 12.

Reviewer 3 Report

Dear Authors,

The topic is interesting, and the manuscript contains fair number of experiments. This manuscript can be worthy of publication after some explanations:

- The technology is called a “near-liquidus squeeze casting”, so in order to perform this process the melting temperature of cast material (alloy) should be precisely know. Therefore, a DTA or DSC measurement is recommended to clarify the melting and solidification temperature of this alloy at normal conditions.

- Page 2: „...the alloys were prepared by remelting commercial AZ91D alloy, ...“ – At what temperature was that remelting done and in what type of furnace?

Page 7, Fig. 9: Tensile strength trials: How many samples were tested for each applied pressure? I think that at least three samples of same category (same applied pressure) should be tested to obtain fair results about its UTS. Also, standard deviation should be included for each batch of tests.

Moreover, authors ignored the yield strength (YS) or proof stress Rp0.2. It is an amount of stress that will result in a plastic strain of 0.2%. Fig. 9 suggests that sample with applied pressure of 170MPa will probably have the highest YS 0.2. For many engineering applications is YS even more important than UTS. Therefore, I recommend including such information in this study.

On page 9 is stated that primary alpha-Mg phase in AZ91D alloy is formed at the liquidus temperature and in Tab.3 is the solidification temperature of this alloy at different applied pressures. However, the solidification process can be greatly influenced by supercooling. How the authors can be sure about the solidification temperatures determined so precisely for atmospheric and different applied pressures (Tab. 3 and 4)? I would advise to include here a cooling curves from K-type thermocouple to support these statements.

English is good, but some typos can be found in the manuscript. For example: page 7, firs sentence in 3.5: “induceed”.

Author Response

1. The technology is called a “near-liquidus squeeze casting”, so in order to perform this process the melting temperature of cast material (alloy) should be precisely know. Therefore, a DTA or DSC measurement is recommended to clarify the melting and solidification temperature of this alloy at normal conditions.

Response:I'm very grateful for your professional opinion. Under the guidance of your professional opinions, we added the DSC in Figure 11 to make it clearer.

2. Page 2: „...the alloys were prepared by remelting commercial AZ91D alloy, ...“ – At what temperature was that remelting done and in what type of furnace?

Response:Thank you very much for your careful review of our manuscript, we have supplemented the casting details of the alloy.

3. Page 7, Fig. 9: Tensile strength trials: How many samples were tested for each applied pressure? I think that at least three samples of same category (same applied pressure) should be tested to obtain fair results about its UTS. Also, standard deviation should be included for each batch of tests.

Response: Thank you for carefully revising our manuscript and your professional comments. Three samples from the sample were selected for mechanical analysis, but the difference between the three is very small. Therefore, I would say this stretch curve is symbolic.

4. Moreover, authors ignored the yield strength (YS) or proof stress Rp0.2. It is an amount of stress that will result in a plastic strain of 0.2%. Fig. 9 suggests that sample with applied pressure of 170MPa will probably have the highest YS 0.2. For many engineering applications is YS even more important than UTS. Therefore, I recommend including such information in this study.

Response: Thanks for your professional advice. This study is based on the automotive differential bracket, which has been described in the introduction. The UTS is more important for such pieces than the YS. Therefore, we neglected to investigate the laws of YS.

5. On page 9 is stated that primary alpha-Mg phase in AZ91D alloy is formed at the liquidus temperature and in Tab.3 is the solidification temperature of this alloy at different applied pressures. However, the solidification process can be greatly influenced by supercooling. How the authors can be sure about the solidification temperatures determined so precisely for atmospheric and different applied pressures (Tab. 3 and 4)? I would advise to include here a cooling curves from K-type thermocouple to support these statements.

Response: Thank you for your professional advice. We have added the DSC curves of the warming process (Figure 11a), which supports our view in a certain extent. Further addition of curves for the cooling process is not necessary.

6. Comments on the Quality of English Language

English is good, but some typos can be found in the manuscript. For example: page 7, firs sentence in 3.5: “induceed”

Response:Thank you for your careful revision of our manuscript and for your professional comments. I have corrected this part in the revised manuscript and highlighted it in red.

Reviewer 4 Report

Review Report

The authors presented an article on “Effect of near-liquidus squeeze casting pressure on microstructure and mechanical property of AZ91D alloy differential support”. The subject of the article falls within the scope of the journal "Materials". However, the article will be ready for publication after a major revision. Comments are listed below.

1.      The introduction is very short. It should be expanded. In addition, the references given in the introduction are insufficient. References should be increased.

2.      In the last paragraph of the introduction, the study's main purpose, its difference from previous studies, and its originality should be stated.

3.      Why was a mixture of CO2 and SF6 used as a protective atmosphere gas? There are other shielding gases as well.

4.      The Materials and methods section should give more details about the tensile test. For example, what are the tensile test parameters? According to what standards was the tensile test performed? In addition, the devices used in microstructure analysis were not mentioned.

5.      In Figure 5, the grain size decreases with the increase in pressure. The reasons for this should be discussed.

6.      The SEM images given in Figure 6 are not clear. The resolution can be increased.

7.      Why does the tensile strength decrease after 140 MPa compression in Figure 9? It should be explained.

8.      The article contains numerous typographic and language errors. It should be corrected.

9.      The article should be rearranged by taking into account the journal writing rules and citation rules.

Author Response

1. The introduction is very short. It should be expanded. In addition, the references given in the introduction are insufficient. References should be increased.

Response:Thank you very much for your reminder, which is very important for revising our manuscript. For this problem, I have corrected it in the manuscript and highlighted it in red. But beyond that, as a postscript, the main referential literature comprising the above-mentioned content in relevance is listed.

2. In the last paragraph of the introduction, the study's main purpose, its difference from previous studies, and its originality should be stated.

Response:Thank you for your guidance. We have added the purpose and novelty of this work in the introduction section.

3. Why was a mixture of CO2 and SF6 used as a protective atmosphere gas? There are other shielding gases as well.

Response:Using mixed gas as protective gas (CO2 and SF6) to prevent oxidation and combustion of liquid magnesium alloys, which have a good insulation effect on the air.

4. The Materials and methods section should give more details about the tensile test. For example, what are the tensile test parameters? According to what standards was the tensile test performed? In addition, the devices used in microstructure analysis were not mentioned.

Response:Thank you very much for your reminder, which is very important for revising our manuscript. I have added relevant content to the revised manuscript.

5. In Figure 5, the grain size decreases with the increase in pressure. The reasons for this should be discussed.

Response:Thank you very much for your reminder, I have updated the picture in a new version of the manuscript.

6. The SEM images given in Figure 6 are not clear. The resolution can be increased.

Response:Thank you very much for your reminding, I have updated the picture in a new version of the manuscript

7. Why does the tensile strength decrease after 140 MPa compression in Figure 9? It should be explained.

Response: Thank you for your professional advice. We have added a description of this phenomenon in the previous section on micro-organizational discussions (page 7).

“The main reason for this is that when pressure is more than critical pressure, the refined grain effect will decline and turn into promote grain growth.”

8. The article contains numerous typographic and language errors. It should be corrected.

Response:Thank you for your guidance, we have submitted a new version of the manuscript with the language carefully polished.

9.The article should be rearranged by taking into account the journal writing rules and citation rules.

Response:Thank you very much for your reminder, which is very important for the revision of our manuscript. We have addressed the comments raised by the reviewers, and the amendments are highlighted in red in the revised manuscript. We particularly revised some sentences since they were not correct.

Round 2

Reviewer 2 Report

The annotations in Figures 6, 8 and 10 in red color need to be improved (smaller font, other color, or moved outside the image)

Author Response

Response:Thank you very much for your reminder, I have updated the picture in a new version of the manuscript.

Reviewer 3 Report

Dear Authors,

the manuscript was improved and after minor correction can be published.
Page 2, section 2. Materials and methods: „In this paper, the alloys were prepared by remelting commercial AZ91D alloy, with the casting temperature of 740 °C in ,“ – In what? The type of furnace is still not defined.

Author Response

Thank you for your careful revision of our manuscript. I have corrected this part in the revised manuscript and highlighted it in red.

Reviewer 4 Report

Review Report#2

 The authors completed the requested revisions. This article can be accepted for publication in its final form.

Author Response

Thank you for your guidance. We hope that the revision is acceptable, and we look forward to hearing from you regarding our submission.